# Transcriptome Profiling Reveals Key Regulatory Networks for Age–Dependent Vernalization in Welsh Onion (*Allium fistulosum* L.)

**DOI:** 10.3390/ijms252313159

**Published:** 2024-12-07

**Authors:** Yin Liu, Dan Wang, Yu Yuan, Yue Liu, Bingsheng Lv, Haiyan Lv

**Affiliations:** 1College of Horticulture, Jilin Agricultural University, Changchun 130118, China; ly2941571760@163.com; 2College of Horticulture, Qingdao Agricultural University, Qingdao 266109, China; wangdan199910@163.com (D.W.); yuanyudyouxiang@163.com (Y.Y.); liuyue@qau.edu.cn (Y.L.)

**Keywords:** Welsh onion, vernalization, age–dependent, transcriptome profiling, RNA–seq

## Abstract

Plants exhibit diverse pathways to regulate the timing of flowering. Some plant species require a vegetative phase before being able to perceive cold stimuli for the acceleration of flowering through vernalization. This research confirms the correlation between the vernalization process and seedling age in Welsh onions. Findings from two vernalization experiments conducted at different time intervals demonstrate that seedlings must reach a vegetative phase of at least 8 weeks to consistently respond to vernalization. Notably, 8–week–old seedlings subjected to 6 weeks of vernalization displayed the shortest time to bolting, with an average duration of 138.1 days. Transcriptome analysis led to the identification of genes homologous to those in *Arabidopsis thaliana* that regulate flowering. Specifically, *AfisC7G05578* (*CO*), *AfisC2G05881* (*AP1*), *AfisC1G07745* (*FT*), *AfisC1G06473* (*RAP2.7*), and *AfisC2G01843* (*VIM1*) were identified and suggested to have potential significance in age–dependent vernalization in Welsh onions. This study not only presents a rapid vernalization method for Welsh onions but also provides a molecular foundation for understanding the interplay between seedling age and vernalization.

## 1. Introduction

Welsh onion (*Allium fistulosum* L.), a perennial vegetable belonging to the genus *Allium* in the family Amaryllidaceae, is widely distributed in East Asian countries [1]. It is characterized by a distinctive, pungent flavor and is frequently consumed for its pseudostems and leaves. Welsh onions also play a significant role in the prevention of human diseases [2,3,4]. However, Welsh onions are susceptible to cold temperatures during fall and spring planting, which can lead to early bolting and a consequent loss of commercial value. Conversely, this cold–induced flowering can be accelerated during the breeding process.

Winter annuals or perennials usually need to be exposed to a cold winter before they can flower in the following spring, a process known as vernalization [5,6,7,8]. Several genes have been identified to play an important role in the vernalization process in *Arabidopsis thaliana* winter annual germplasm. Before exposure to cold, FRIGIDA (FRI) activates the expression of *FLOWERING LOCUS C* (*FLC*), which represses *FLOWERING LOCUS T* (*FT*) and *SUPPRESSOR OF OVEREXPRESSION OF CONSTANS 1* (*SOC1*) to inhibit flowering [9,10,11,12,13,14,15]. Cold induces the formation of FRI nuclear condensates, which segregate the binding between FRI and *FLC*, thereby inhibiting *FLC* expression [16,17]. At the same time, cold stimulates the epigenetic silencing of *FLC* by Polycomb Repressive Complex 2 (PRC2), which promotes flowering [18,19]. In wheat, *VERNALIZATION 1* (*VRN1*), *VERNALIZATION 2* (*VRN2*), and *VERNALIZATION 3* (*VRN3*) play a major role in vernalization [20,21]. *VRN1* is a flowering activator that is activated in response to cold [22,23]. *VRN2* is a flowering repressor that is activated during long daylight cycles and down–regulated during vernalization [24]. *VRN3* is a homolog of *FT* in *Arabidopsis thaliana* that interprets environmental signals to promote flowering [25]. After prolonged exposure to cold, the up–regulation of *VRN1* expression releases *VRN3* by repressing *VRN2* expression, which further promotes *VRN1* expression and accelerates flowering [5,20,21]. In onions, the *FT* homologous genes, *AcFT1* and *AcFT2*, were cloned, and their overexpression in *Arabidopsis thaliana* accelerated flowering [26].

The vernalization of plants can occur in germinating seeds or young seedlings, and these two types are called seed vernalization type and green vernalization type, respectively. For instance, the seed vernalization type of plant, winter rapeseed (*Brassica rapa*), undergoes vernalization after direct cold treatment of the germinating seeds [27]. However, certain plants require a specific age base for vernalization to occur. For example, *Arabis alpina* and *Cardamine flexuosa* respond to cold exposure after at least five weeks of age [28,29,30,31]. During this process, miR156 expression decreases with age, and *SPLs* are released to activate miR172 expression to ensure vernalization can occur [32]. Simultaneously, cold induces the up–regulation of *VRN1* expression, which enhances miR172 expression and suppresses the expression of the APETALA2 transcription factor (*AP2L*, a flowering repressor that is the target of miR172), releasing *VRN3* to promote flowering [28,29,32,33]. However, the vernalization process still holds unknown effects, and the mechanisms of how plants perceive vernalization remain unclear. In Welsh onions, vernalization is highly age–sensitive in almost all varieties, resulting in a significant difference in flowering time between vernalized and non–vernalized Welsh onions. However, the seedling age threshold for perceiving vernalization in Welsh onions is unclear, and there is no foundation or direction for the molecular mechanisms of age–dependent vernalization in Welsh onions.

Here, we determined that the perception of vernalization in Welsh onions correlates with age and identified the critical conditions for a stable vernalization process. Welsh onions can stably undergo vernalization and accelerate flowering when exposed to cold for 6 weeks at 8 weeks old or for 4 weeks at 10 weeks old. Using the genome of the bunching onion as a reference [34,35], we analyzed the transcriptome during this process using RNA–seq to examine differentially expressed genes in the vernalization pathway versus the age pathway over time [36]. This study provides a set of reliable tools for the accelerated breeding of Welsh onions and lays a solid foundation for studying the molecular mechanisms of vernalization in Welsh onions.

## 2. Results

### 2.1. Welsh Onions Require at Least an 8–Week–Old Age Foundation to Stably Perceive Vernalization

Seedling age and cold duration are important factors affecting vernalization in Welsh onions. To investigate the critical seedling age and cold duration required for Welsh onion vernalization, cold treatments were conducted at different seedling ages. The results showed a significant correlation between Welsh onion seedling age and the impact of cold vernalization. Welsh onions need more than 8 weeks of vegetative growth to reliably undergo vernalization (Figure 1A). Two types of vernalization trials, labeled as Group A and Group B, were carried out at different time points, and observations were recorded for up to 200 days. In the Group A trial, Welsh onions grown to 5 weeks old under room–temperature conditions did not bolt when subjected to 3 (N5V3) weeks of cold treatment. With a 6–week (N5V6) cold treatment, only one Welsh onion bolted, and with a 9–week (N5V9) treatment, only two Welsh onions bolted. On the other hand, when Welsh onions were grown to 8 (N8V3) or 11 (N11V3) weeks, a 3–week cold treatment did not cause all Welsh onions to bolt within the specified time. However, the bolting rate tended to increase with age, and with 6 (N8V6, N11V6) or 9 (N8V9, N11V9) weeks of cold treatment, all Welsh onions bolted, indicating successful vernalization (Figure 1B). These findings suggest that Welsh onions face challenges in passing vernalization at younger seedling ages, with the vernalization requirement decreasing as the seedlings mature. The Group B trial confirmed the results of Group A. In this trial, a 4–week low–temperature vernalization treatment when Welsh onions were 4 weeks old did not induce bolting within the specified time. When the onions were 10 weeks old, cold treatments of 2 weeks or less failed to meet vernalization requirements, while a 4–week treatment allowed the onions to pass vernalization reliably (Figure 1C). These results show that the vernalization process in Welsh onions is strictly limited by the age of the seedlings, with the critical age threshold being 8 weeks old, beyond which, Welsh onions can stably perceive vernalization.

We counted the time to bolting for Welsh onions in each treatment and found that when vernalization treatments were performed at the same seedling age, the time to recover and bolt decreased with the increasing duration of cold treatment. However, for the total time to bolting, a prolonged period of cold treatment after vernalization requirements were met rather delayed the time to bolting (Figure 1D). Here, we determined that Welsh onions exposed to cold for 6 or more weeks at 8 weeks old or more, or 4 weeks at 10 weeks old, could satisfy Welsh onion’s demand for vernalization and that the total bolting time for fully vernalized Welsh onions was shortest in the N8V6 treatment, averaging only 138.1 days (Figure 1D). Nevertheless, the Welsh onion bolting time appeared to be much more stable in the N10V4 treatment condition, with an average bolting time of only 141.4 days (Figure 1E).

### 2.2. Transcriptome Analysis of Pseudostems and Stem Disks During Vernalization of Welsh Onions

To analyze the effects of seedling age and duration of cold on the transcriptome of Welsh onions during vernalization, we sampled pseudostems and stem disks before and after vernalization treatments. Corresponding non–vernalized Welsh onion parts were collected in three biological replicates. Based on the results of the vernalization test, in Group A, we selected the vernalized non–bolting sample N5V3, the partial–bolting sample N8V3, and the complete–bolting sample N8V6, along with non–vernalized samples before and after each vernalization treatment (N5V0, N8V0, N11V0, and N14V0) for transcriptome construction. Furthermore, in Group B, we chose the vernalized non–bolting samples N4V4, the non–bolting sample N10V1, and the complete–bolting sample N10V4, as well as non–vernalized samples before and after each vernalization treatment (N4V0, N8V0, N10V0, N11V0, and N14V0) for transcriptome construction. A total of 45 samples, including 21 samples in Group A and 24 samples in Group B, were collected for RNA–seq assays. High–quality (Q30 > 95%) clean reads of 971,623,414 and 1,080,472,122 were obtained for Group A and Group B, respectively, with an average of 46,267,782 and 45,019,672 clean reads. These clean reads were mapped to the bunching onion genome [34], with a mapping ratio exceeding 95% (Appendix A). Correlation heatmaps and principal component analyses (PCAs) based on three biological replicates of samples from different treatments showed a significant correlation between samples from different treatments after excluding several obvious outliers, with notable differences between vernalized and non–vernalized samples, particularly in Group A (Figure 2).

Transcriptome expression profiles revealed the genes expressed in the samples from different treatments. In Group A, 37,578, 36,510, 36,555, 36,506, 36,407, 34,401, and 34,736 genes were expressed during the N5V0, N8V0, N11V0, N14V0, N5V3, N8V3, and N8V6 periods, respectively. It was found that 30,030 genes were co–expressed in all treatments (Figure 2E). In Group B, during the N4V0, N8V0, N10V0, N11V0, N14V0, N4V4, N10V1, and N10V4 periods, 38,164, 35,531, 36,655, 36,516, 35,841, 36,834, 35,672, and 35,485 genes were expressed, of which, 29,769 genes were found to be co–expressed in all treatments (Figure 2F).

### 2.3. Expression Cluster Analysis

To analyze the main dynamics during vernalization, a time series trend analysis was conducted for all expressed genes. Based on the clustering of genes with similar expression profiles, all expressed genes were classified into 10 different clusters (C1–C10). Among the expression patterns in Groups A and B (Figure 3 and Appendix A), the expression patterns of the 10 clusters in Group B were more comprehensive and could better demonstrate the main dynamics of genes during vernalization. In Group B, clusters C1–C10 contained 3877, 4714, 6725, 3523, 4244, 5667, 4000, 2984, 3984, and 3607 genes, respectively. Among these clusters, 1, 4, 5, 7, 8, 9, and 10 showed only one peak in N8V0, N10V1, N14V0, N4V0, N10V0, N4V4, and N11V0, respectively, and maintained the same level in the other treatments. The time series trend of these clusters deviated from the expected trend.

Additionally, the time series tendencies of cluster 2, cluster 3, and cluster 6 were strongly correlated with the expected dynamic trends during vernalization. Gene expression levels in cluster 2 were higher in treatments that did not undergo vernalization and decreased rapidly after vernalization. These genes were primarily associated with RNA modification and processing. Cluster 3 maintained the same level in other treatments but was rapidly elevated in N10V4, which passed through vernalization completely. The genes in this cluster were mainly enriched in protein ubiquitination and proteolytic–related processes. Cluster 6 had higher levels in the younger–seedling–age Welsh onion treatments and lower levels in the older–seedling–age treatments. The genes in this cluster may play an important role in the age pathway regulation of flowering. The genes in this cluster were mainly enriched in the nuclear–transcribed mRNA catabolic process, nonsense–mediated decay, translational initiation, virus transcription, SRP–dependent cotranslational protein targeting to membrane, and rRNA processing (Figure 3). Overall, these 10 clusters were able to distinguish the expression profiles of all genes. These expression patterns indicate the main relationships and differences in response to seedling age and vernalization interactions.

### 2.4. Analysis of Cold–Induced Differential Genes

To understand the molecular differences during cold treatment, each cold–treated sample was screened for differential genes compared to the corresponding normal temperature samples and pre–treatment samples. In Group A, 1171, 1636, and 1488 up–regulated differential genes were identified in Cold_N5V3, Cold_N8V3, and Cold_N8V6, respectively. Simultaneously, 1159, 1935, and 1715 down–regulated differential genes were identified, respectively (Figure 4A). In Group B, 762, 1056, and 2057 up–regulated differential genes were identified in Cold_N4V4, Cold_N10V1, and Cold_N10V4, and 1089, 931, and 2791 down–regulated differential genes were identified, respectively (Figure 4B). These differential genes underwent GO enrichment analysis, revealing that the main enriched biological processes shared high similarity. These commonly enriched biological processes may play a crucial role in the onion’s acclimation to cold and adaptation to photoperiodic changes, particularly in response to high light intensity, hydrogen peroxide, the regulation of chlorophyll biosynthetic process, and the positive regulation of seed germination (Figure 4C).

### 2.5. Differential Gene Analysis with Age

To investigate the molecular differences in the age–regulated vernalization process of Welsh onions, we screened for candidate genes involved in this process. We compared the old–seedling–age samples that underwent vernalization treatment with the young–seedling–age samples, and similarly compared the old–seedling–age samples in the vernalization treatment that did not experience vernalization with the young–seedling–age samples. In Group A, we identified the overlap in differential genes between N11V0, N14V0, and N5V0, as well as between N8V3, N8V6, and N5V3, which included 13 up–regulated genes and 61 down–regulated genes (Figure 5A). In Group B, the overlap of differential genes between N11V0, N14V0, and N5V0 and between N10V1, N10V4, and N4V4 resulted in 116 up–regulated genes and 789 down–regulated genes (Figure 5B). Subsequently, to identify genes with stable expression across the two groups, we performed expression cluster analysis within each group. Group A showed no significant up–regulation clusters, while clusters 6 to 10 all exhibited down–regulation, aligning with the trends observed in clusters 1 to 4 and clusters 6 to 10 in Group B (Figure 5C,D). From these analyses, we selected differential genes with consistent expression trends in both groups for GO enrichment analysis.

The up– and down–regulated differential genes in Group A and Group B were separately overlapped, resulting in the identification of 688 commonly down–regulated genes (Figure 6A). A total of 113 biological processes were enriched through GO enrichment analysis. These groups of expressed genes were enriched in biological processes related to nucleosome assembly, epigenetic regulation, and gibberellin signaling, as well as in areas of meristem maintenance, the regulation of growth and transition from the vegetative to the reproductive phase of meristem, and pathways related to meristem organization regulation and flowering (Figure 6B). Eight *Arabidopsis thaliana* homologous genes involved in the regulation of flowering were screened, including four *VIM1* homologous genes, *AfisC1G06103*, *AfisC2G01843*, *AfisC2G06661*, and *AfisC7G03599*; two *RAP2.7* homologous genes, *AfisC1G06473* and *AfisC5G04457*; the *ATH1* homologous gene *AfisC2G04220*; and the *AHL22* homologous gene *AfisC3G06612*. These genes negatively regulate flowering in *Arabidopsis thaliana* [37,38,39,40] and, as expected, were highly expressed in vernalization–treated versus non–vernalized samples at younger seedling ages, with a tendency to be down–regulated at older seedling ages (Figure 6C,D).

### 2.6. Analysis of Differential Genes During Vernalization

To investigate the molecular differences during cold–induced vernalization in Welsh onions, we screened for candidate genes associated with vernalization. Samples that underwent vernalization were compared with adjacent untreated samples and samples that did not undergo vernalization. In Group A, the up–regulated and down–regulated differential genes of N8V6_VS_N8V0, N8V6_VS_N14V0, and N8V6_VS_N5V3 were overlapped, resulting in 348 up–regulated and 388 down–regulated differential genes, respectively (Figure 7A). In Group B, the up–regulated and down–regulated differential genes of N10V4_VS_N10V0, N10V4_VS_N14V0, N10V4_VS_N4V4, and N10V4_VS_N10V1 were overlapped separately, containing 465 up–regulated and 908 down–regulated differential genes, respectively (Figure 7B). All differential genes screened in both groups were analyzed for expression clusters in Group A and Group B. In Group A, clusters 7 and 9 were up–regulated in N8V6, and clusters 1, 3, 6, and 10 were down–regulated in N8V6 (Figure 7C). In Group B, clusters 4, 5, and 7 were up–regulated in N10V4, while clusters 1, 6, 8, 9, and 10 were down–regulated in N10V4 (Figure 7D). Genes with stable expression in Group A and Group B were screened, and these genes were analyzed for GO enrichment.

The up– and down–regulated differential genes in Group A and Group B were overlapped separately, and 308 genes were screened for common up–regulation, while 619 genes were identified for common down–regulation (Figure 8A). GO enrichment identified a total of 31 biological processes, with these genes primarily enriched in processes such as viral transcription, SRP–dependent cotranslational protein targeting to the membrane, nuclear–transcribed mRNA catabolic process, nonsense–mediated decay, as well as in the regulation of meristem structural organization, secondary bolting formation, the regulation of floral meristem growth, and rhythmic processes, which were significantly enriched (Figure 8B). Five *Arabidopsis thaliana* genes homologous to those regulating flowering were identified, *AfisC7G05578* (*CO*), *AfisC5G00860* (*COL1*), *AfisC2G05881* (*AP1*), *AfisC1G07745* (*FT*), and *AfisC2G01843* (*VIM1*). *COL1* is a homologous gene of *CO*, and *AP1*, *FT*, and *CO* are promoters of flowering [13,41,42,43,44]. As expected, they were significantly up–regulated during vernalization. The inhibition of flowering upon the overexpression of *VIM1* was significantly down–regulated in untreated old–seedling–age samples and old–seedling–age vernalization–treated samples, and it was lowest in samples that had passed vernalization (Figure 8C,D), suggesting that the *VIM1* inhibition of flowering may be related to both vernalization and age [37].

### 2.7. Weighted Correlation Network Analysis (WGCNA) and Visualization

To explore the correlation network between the age pathway and vernalization, the WGCNA technique was employed to identify modules related to the age pathway regulation of vernalization. Expressed genes from N4V0, N4V4, and N10V4 samples in Group B, which were highly correlated with the age pathway in regulating vernalization, were selected to construct the WGCNA co–expression network. Similar branches of the gene clustering tree were merged (Figure 9A), resulting in a total of 48 co–expression modules, with the largest proportion of genes found in the blue module. The correlations in the age pathway regulation of vernalization were similar for N4V0 and N4V4 but were opposite for N10V4. Among all modules, only the blue module was consistent with expectations. Meanwhile, the candidate genes *AfisC1G06103*, *AfisC1G06473*, *AfisC2G01843*, *AfisC2G04220*, *AfisC2G06661*, *AfisC3G06612*, and *AfisC7G03599* in age–regulated vernalization, as well as the candidate gene *AfisC7G05578* in vernalization, are all located in the blue module (Figure 6, Figure 8 and Figure 9B).

To further investigate the key regulatory networks involved in the age pathway regulation of vernalization, candidate flowering regulatory genes and all transcription factors from the age process and vernalization process were included in the constructed correlation network within the blue module (Figure 6, Figure 8 and Figure 9). A total of 84 genes with a weight value greater than or equal to 0.4 were analyzed. Among these, 31 genes had a linkage of less than 10 and were positioned in the outermost circle. Twenty–six genes had a linkage of 20 or greater and were situated in the middle two circles. Only eight genes had a connectivity of 40 or higher and were located in the central circle. This group included two *VIM1* candidates, as well as C3H (*AfisC1G05806*), GeBP (*AfisC8G05944*), AP2/ERF–AP2 (*AfisC5G00393*), bHLH (*AfisC4G02772*), WRKY (*AfisC2G07934*), and C2H2 (*AfisC2G07291*) family transcription factors, which may act as key switches in the age–regulated vernalization of Welsh onion (Figure 10).

## 3. Discussion

Previous studies have shown that there is an age limitation on vernalization in many plants. For example, *Arabis alpina* and *Cardamine flexuosa* need to be at least five weeks old before vernalization can proceed stably. This process is related to both the age pathway and the vernalization pathway [28,29]. In Welsh onions, we observed similar results (Figure 1). However, the molecular mechanism underlying this process in Welsh onions is unknown, and such studies are rare within the genus *Allium*. In this study, we obtained the transcriptional profiles of Welsh onions during vernalization using RNA–seq. We constructed differential gene modules for both the age pathway and the vernalization pathway based on the expression patterns of genes in these pathways, respectively. This approach enhanced our molecular understanding of the vernalization aspect in Welsh onions.

### 3.1. Age Basis of Vernalization in Welsh Onion

Vernalization plays a crucial role in the life cycle of many plants [5,6,7,8,45]. For certain species, such as *Arabis alpina* and *Cardamine flexuosa*, the ability to perceive vernalization is contingent upon attaining a specific seedling age. Welsh onions also exhibit an age–dependent sensitivity to vernalization, but the exact age at which they become responsive was previously unknown. Our experiments have now elucidated the critical conditions for Welsh onions to perceive vernalization. Notably, Welsh onions over 8 weeks old can rapidly complete vernalization with an appropriate duration of low temperatures, as demonstrated by the N8V6 and N10V4 parameters. This behavior mirrors that of other plants with age sensitivity [28,29], suggesting that by the 8–week seedling stage, Welsh onions have reached the necessary age threshold for vernalization perception.

Curiously, when we exposed 5–week–old Welsh onions to extended periods of cold, we observed that a minority were capable of sensing and responding to vernalization, as indicated by the N5V6 group. However, prolonging their exposure to low temperatures only resulted in a modest increase in the bolting rate, as seen in the N5V9 group. This indicates that at the 5–week stage, Welsh onions have not yet met the age requirement for vernalization perception. When observing these treatments, it becomes evident that even though the plants had not yet reached the optimal seedling age at the onset of vernalization, the prolonged exposure to low temperatures may have allowed some of them to eventually attain the necessary age threshold for sensing cold, thereby successfully completing the vernalization process.

### 3.2. Identification of Candidate Genes in the Vernalization Process

Vernalization and bolting in plants are intricately regulated by a multitude of factors, with Welsh onions exemplifying the involvement of both age–dependent and vernalization pathways. By leveraging the outcomes of vernalization experiments and the homology with *Arabidopsis* flowering regulatory genes, we have identified numerous differential genes and several candidate genes that align with the anticipated functional patterns. Notably, genes such as *AfisC1G07745* (*FT*) [13,25], *AfisC7G05578* (*CO*), and *AfisC5G00860* (*COL1*) [41,42] were significantly up–regulated under N8V6 and N10V4 conditions, underscoring their pivotal roles in the vernalization process of Welsh onions [44,46]. Additionally, genes like *AfisC2G05881* (*AP1*) [43,47,48,49] and *AfisC2G01843* (*VIM1*) [37,50,51], while not perfectly aligning with their *Arabidopsis* counterparts, exhibit broad consistency and are thus considered candidate genes. However, key genes that are crucial in the vernalization process of most plants, including *FLC*, *FRI*, *VRN1*, and *VRN2*, did not reveal homologs in our RNA–seq analysis. This absence may suggest that Welsh onions possess unique characteristics, potentially due to unknown processes that interfere with the expression of these genes or to specific evolutionary events that have led to significant divergence, complicating their homologous identification. Nonetheless, we are confident that these genes are likely included among the differentially expressed genes we have identified.

### 3.3. The Functional Relationships of Candidate Genes in the Vernalization of Welsh Onions

Plants’ perception of vernalization and their needs at different stages involve many complex and precise regulatory patterns, which are truly astonishing [52]. In our study, we have screened numerous candidate genes and transcription factors, but the exact regulatory patterns still require further experimentation to verify. Here, we refer to some existing studies to provide a general direction for their relationships. The AP2 family transcription factor *RAP2.7* (related to *AP2.7*; *TOE1*) is a target of miR172 in the age pathway and acts as a negative regulator of flowering [53,54]. *RAP2.7* interacts physically with *CO* proteins to inhibit *FT* by suppressing *CO*, playing a crucial role in age– and photoperiod–mediated vernalization, which decreases with age [40,54,55,56]. In this research, two *RAP2.7* homologs, *AfisC1G06473* and *AfisC5G04457*, decreased as seedling age increased, both in vernalization and non–vernalization. *ATH1* encodes a transcription factor involved in photomorphogenesis, regulates gibberellin biosynthesis, and is activated by AGAMOUS in a cal–1, ap1–1 background, acting as an *FLC* activator that significantly delays flowering in C24 [38,57,58]. *AHL22* is a chromatin remodeling factor that negatively regulates flowering through histone 3 (H3) methylation and acetylation modifications of *FT* chromatin [39,59]. The *VIM1* protein is involved in the regulation of DNA methylation and histone modification status to control genome–wide epigenetic gene silencing, delaying flowering upon overexpression [37,50,51]. In this study, the *RAP2.7* homologous genes *AfisC1G06473* and *AfisC5G04457* were down–regulated with age in Welsh onion, while there was a gradual up–regulation with age in the springing of the *CO*, *COL1*, and *FT* homologous genes *AfisC7G05578*, *AfisC5G00860*, and *AfisC1G07745,* respectively. This suggests that the process of flowering inhibition by *RAP2.7* in *Arabidopsis thaliana* is also present and redundant in Welsh onion. It is critical to point out that genes down–regulated with age, including the *VIM1* homologous genes *AfisC1G06103*, *AfisC2G01843*, *AfisC2G06661*, and *AfisC7G03599*, the *RAP2.7* homologous genes *AfisC1G06473* and *AfisC5G04457*, the *ATH1* homologous gene *AfisC2G04220*, and the *AHL22* homologous gene *AfisC3G06612*, are likely involved in the regulation of the timing of bolting in Welsh onions.

## 4. Materials and Methods

### 4.1. Experimental Design and Sampling

The Welsh onion used in the experiment was ‘Zhangqiu Dawutong’, provided by the Vegetable Functional Gene Innovation Team of Qingdao Agricultural University. Welsh onion seeds were evenly sown in 50–hole cavity trays (Taizhou Longji Plastic Co., Ltd., Taizhou, China) and placed indoors (23 ± 2 °C all day, humidity level of 55–65%, and a 16 h photoperiod) for three weeks, growing to the 1–2 leaf stage. Subsequently, Welsh onion seedlings with normal and uniform growth were transplanted into 15–hole cavity trays for further growth. In the Group A trial, when Welsh onions reached 5, 8, and 11 weeks old, they were transferred to a cold light incubator (Ningbo Kesheng Experimental Instrument Co., Ltd., Ningbo, China) (15–5 °C from day to night, humidity level of 55–65%, and a 12 h photoperiod) for 3, 6, and 9 weeks of vernalization treatment, respectively (Figure 1B,D). More than nine Welsh onions were left untreated as a control. In the Group B experiment, when Welsh onions were 4 weeks old, they were vernalized for 4 weeks. When Welsh onions reached 10 weeks old, they underwent vernalization treatment for 1, 2, and 4 weeks, respectively (Figure 1C,E). More than nine Welsh onions were left untreated as a control. At the end of each vernalization treatment, the Welsh onions were transferred to indoor conditions for recovery and bolting, and bolting rates were recorded for up to 200 days. The results of the vernalization treatments were used to determine the rate of bolting. Three biological replicate samples were harvested before the start of vernalization treatment, after the end of vernalization treatment, and by the age of seedlings without vernalization treatment. The sampling site was the stem disk and pseudostem, with one sample being a mix of every three or more plants (meeting the sequencing standard). The samples were frozen with liquid nitrogen for 2 min after sampling and stored in a refrigerator at −80 °C. Subsequently, the remaining Welsh onions were moved indoors for recovery and bolting.

### 4.2. RNA Extraction and RNA–Seq Library Construction

RNA–seq samples were frozen in dry ice and sent to Annoroad Co., Ltd. (Beijing, China) for RNA extraction and transcriptome sequencing. The product type selected was a common eukaryotic reference transcriptome, and the library type chosen was a eukaryotic transcriptome plus a eukaryotic strand–specific transcriptome. The library was constructed and sequenced using the DNBSEQ–T7 sequencing platform based on DNBSEQTM technology.

### 4.3. Identification of Differential Genes

After downstream data processing, the sequencing results returned raw data containing Rawdata and Cleandata (after removing splice sequences, null reads, and low–quality reads). Cleandata were then compared to the bunching onion genome [34] using HISAT2 v2.2.1 [60]. SAMtools v1.6, Sambamba v0.8.2, and StringTie v2.2.1 [61,62,63] were utilized to obtain positional information on the reference genome. Gene expression levels were calculated for each sample, and gene expression identification was conducted using FPKM. The comparison data were transformed into a full gene expression matrix for sample clustering heatmap analysis and PCA clustering analysis [64]. The samples were graphically analyzed using ggplot2 v3.5.0 [65]. Data from significantly deviating samples were excluded, and the gene expression matrix was reconstructed by calculating the mean between each treatment for the remaining samples. Differential gene calculations were performed using DESeq2 v1.41.10 [66], with a screening range of |log2FoldChange| > 1 and a *p*–value < 0.05. Venn diagram analysis was performed online in BioLadder “https://www.bioladder.cn/ (accessed on 5 June 2024, to 25 June 2024 )” [67].

### 4.4. Gene Clustering and GO Enrichment

ClusterGVis v0.1.1 [68] was utilized for gene clustering based on “mfuzz“ [69]. GO enrichment analysis was conducted using the Functional Annotation Library of Fractionated Welsh onion Genes in the AlliumDB [35]. The GO enrichment analysis was performed with clusterProfiler v4.10.0 [70] to refine and organize the enriched terms with a *p*–value of 0.05 and q–value of ≤0.2. The p–adjust method chosen was the “BH” method, and the GO–enriched term bubble plots were generated using ggplot2 [65].

### 4.5. Weighted Correlation Network Analysis (WGCNA)

The analysis of RNA–seq data and the construction of gene co–expression networks were performed using the R package WGCNA v1.72.5 [71]. Gene networks were generated based on the mean FPKM of gene expression (average FPKM of replicate samples across treatments). Neighbor–joining matrices were constructed using an optimal soft threshold, and networks were identified using the dynamic hybrid tree cut algorithm with a minimum cluster size of 30 and a merge threshold of 0.25. The networks were visualized using Cytoscape v3.10.1 for visualization purposes.

## 5. Conclusions

Our study confirms that Welsh onions require at least an 8–week age foundation to perceive vernalization, providing two reliable vernalization schemes, N8V6 and N10V4, which can bolt in just 138.1 days and 141.4 days, respectively. Concurrently, through transcriptome analysis, we have identified numerous differential genes and further confirmed several candidate genes, such as *AfisC7G05578* (*CO*), *AfisC5G00860* (*COL1*), *AfisC1G07745* (*FT*), *AfisC1G06473* (*RAP2.7*), *AfisC1G06103* (*VIM1*), *AfisC2G01843* (*VIM1*), and *AfisC2G04220* (*ATH1*), among others. The expression trends of these genes are consistent with their functional patterns in *Arabidopsis thaliana*, suggesting that they are highly likely to serve similar functions in Welsh onions. These findings lay a molecular foundation for understanding the interaction between seedling age and vernalization in the Welsh onion.

## Figures and Tables

**Figure 1 ijms-25-13159-f001:**
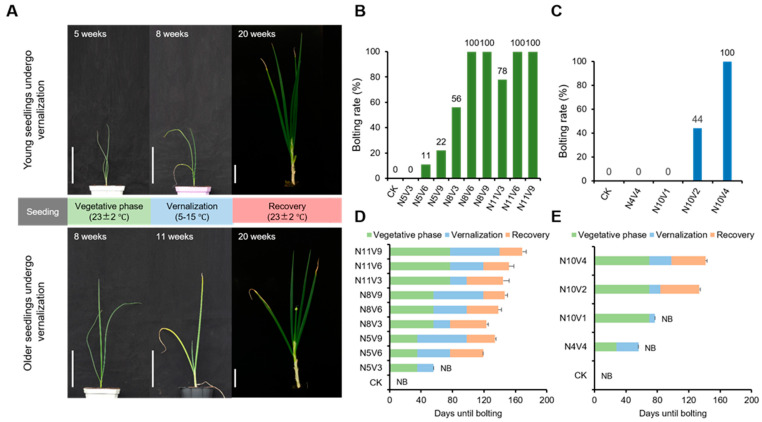
The figure shows the relationship between seedling age and vernalization in Welsh onions. (**A**) The upper panel depicts Welsh onions grown under normal temperature for up to 5 weeks, those grown at normal temperature for 5 weeks followed by a 3–week cold treatment (total growth period of 8 weeks), and their appearance at the end of the cold treatment with a total growth period of 20 weeks. The lower panels display Welsh onions grown for 8 weeks at normal temperature, those grown for 8 weeks at normal temperature followed by a 3–week cold treatment (11–week total growth period), and their appearance at the end of the cold treatment with a total growth period of 20 weeks. Scale bar: 10 cm. Bolting rates for treatments in Groups A (**B**) and B (**C**) were monitored up to day 200 of the entire growth period. The mean time to bolting for each treatment in Groups A (**D**) and B (**E**) was observed up to day 200 of the total growth period, excluding non–bolting shallots. NB indicates no bolting. N5V0 indicates growth at normal temperature up to 5 weeks of age, followed by no vernalization; N5V3 indicates growth at normal temperature up to 5 weeks of age, followed by 3 weeks of vernalization, and so on.

**Figure 2 ijms-25-13159-f002:**
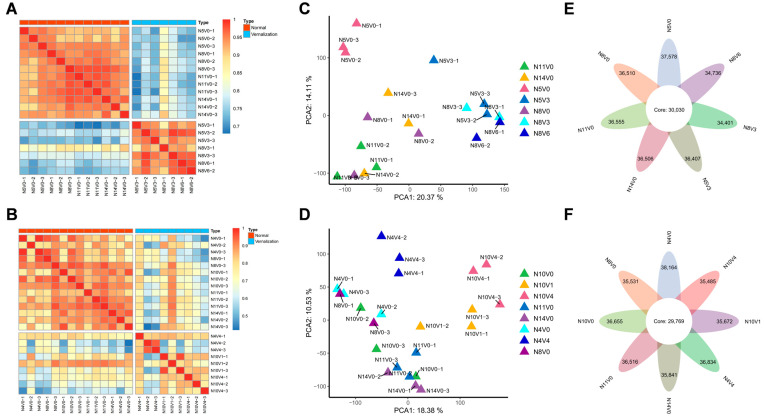
Sample correlation analysis. Correlation heatmaps depict non–vernalized samples in orange–red and vernalized samples in aqua–blue for both Groups A (**A**) and B (**B**). PCA cluster analyses are presented for samples from Groups A (**C**) and B (**D**), with each color representing a distinct treatment. Expression levels are shown after averaging samples from all treatments in Groups A (**E**) and B (**F**). The outermost circle indicates the number of genes expressed in each treatment, while the core position represents the number of genes common to all treatments.

**Figure 3 ijms-25-13159-f003:**
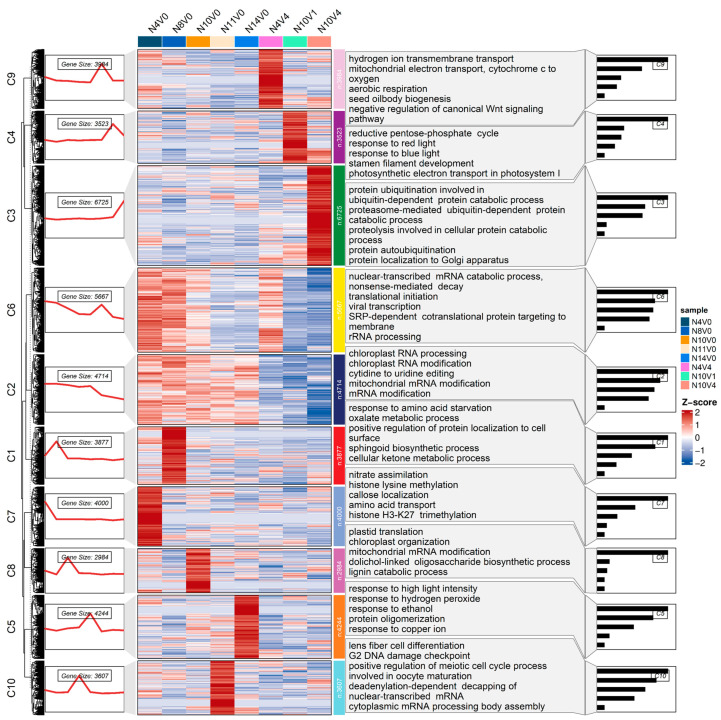
Time series analysis of dynamic gene expression changes during Group B vernalization is depicted in the figure, which presents cluster names, time series plots, all gene expression profiles, major Gene Ontology (GO) enrichment processes (biological processes), and bar graphs from left to right.

**Figure 4 ijms-25-13159-f004:**
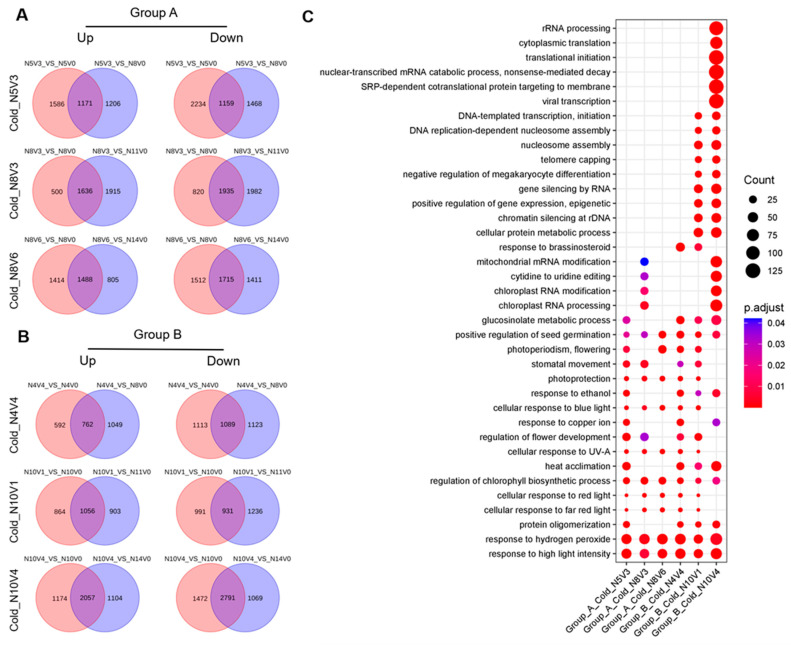
Analysis of differential genes during cold treatment. (**A**) and (**B**) represent the screening process for differential genes during cold treatment in each group, respectively. (**C**) Bubble plots were used to illustrate the GO enrichment analysis of the differential genes identified during cold treatment. The plots highlight the top 10 biological processes enriched in each cold treatment group, along with their co–enrichment entries.

**Figure 5 ijms-25-13159-f005:**
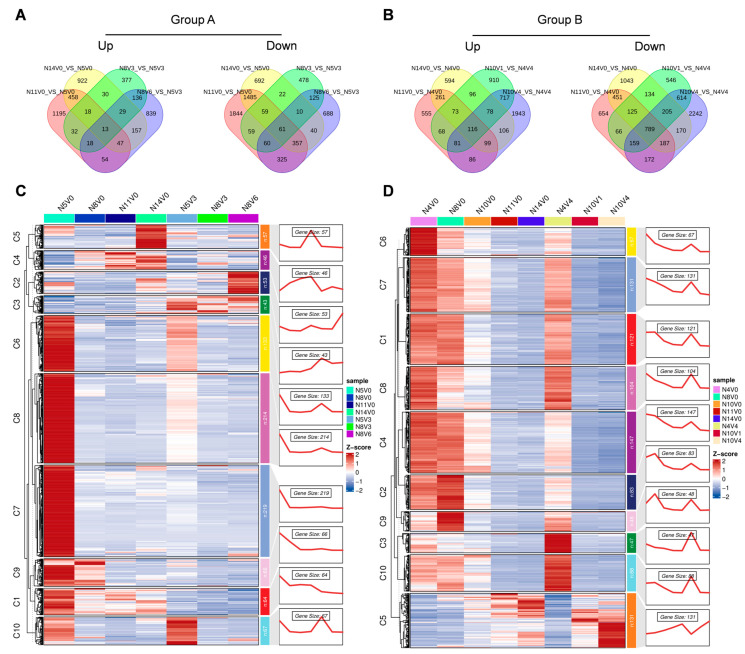
Analysis of differential genes in the aging pathway. Venn diagrams illustrate the common and unique differential genes associated with the aging pathway in Group A (**A**) and Group B (**B**), with age–related differential genes positioned in the central areas. Heatmaps depict the expression patterns of all age–related differential genes in Group A (**C**) and Group B (**D**), respectively.

**Figure 6 ijms-25-13159-f006:**
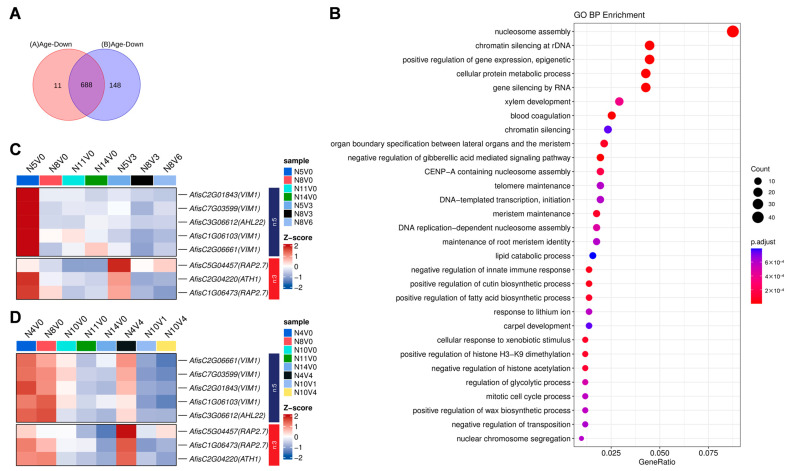
Selection and GO enrichment of differentially expressed genes in the aging pathway. (**A**) Venn diagram illustrates the down–regulated differentially expressed genes during vernalization in Group A and Group B. (**B**) Expression patterns of stable differentially expressed genes were analyzed for GO enrichment, highlighting the top 30 terms. (**C**) and (**D**) Expression patterns of genes regulating homologous genes for *Arabidopsis thaliana* flowering in Groups A and B, respectively.

**Figure 7 ijms-25-13159-f007:**
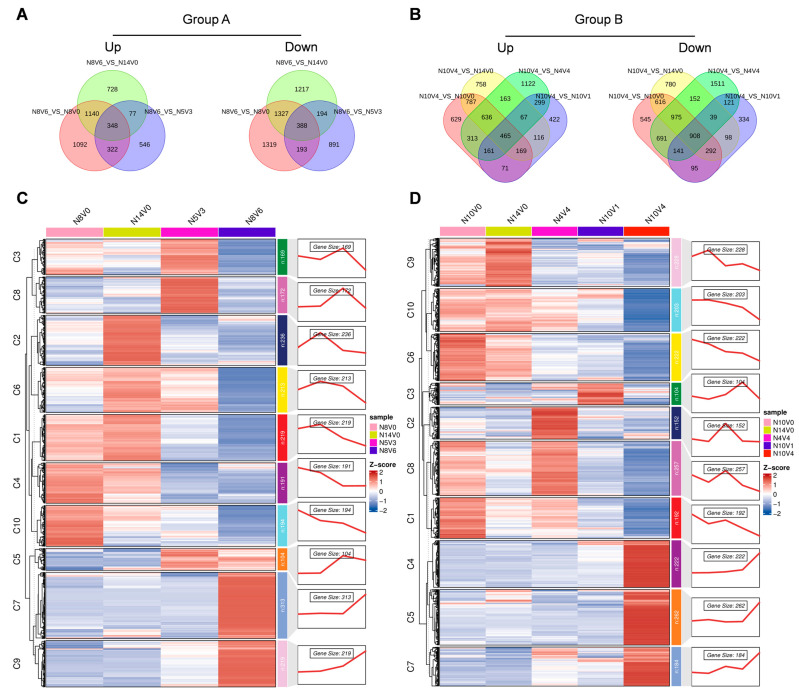
Analysis of differentially expressed genes during cold–induced vernalization. Venn diagrams display the common and unique differentially expressed genes during vernalization in Group A (**A**) and Group B (**B**), with the vernalization–associated differentially expressed genes located in the intersecting sections. Heatmaps depict the expression patterns of all vernalization–associated differentially expressed genes in Group A (**C**) and Group B (**D**), respectively.

**Figure 8 ijms-25-13159-f008:**
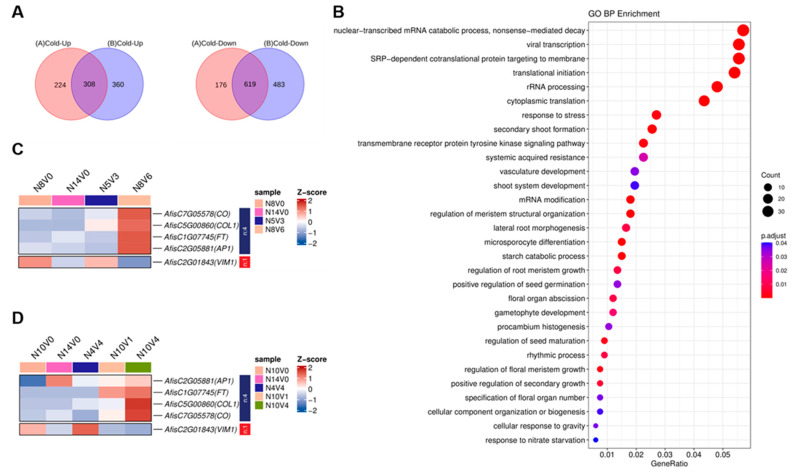
Selection and GO enrichment of differentially expressed genes during cold–induced vernalization. (**A**) The Venn diagram illustrates the overlap of up–regulated and down–regulated differentially expressed genes during vernalization in Group A and Group B. (**B**) The expression patterns of stably differentially expressed genes and GO enrichment analysis are presented, highlighting the top 30 terms. Heatmaps display the expression patterns of genes that regulate homologous genes involved in *Arabidopsis thaliana* flowering in Group A (**C**) and Group B (**D**), respectively.

**Figure 9 ijms-25-13159-f009:**
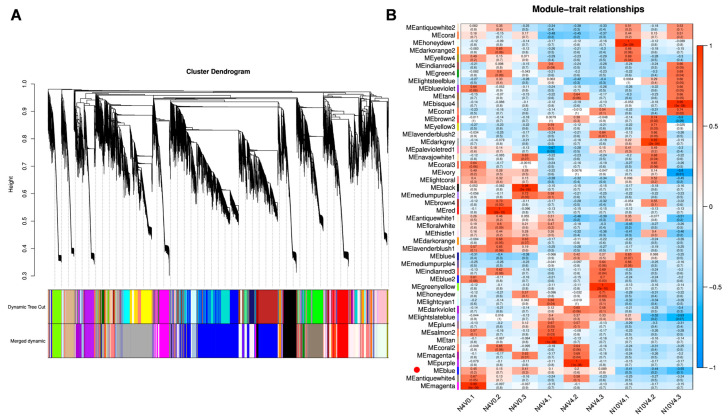
Weighted correlation network analysis. (**A**) Gene clustering and module cutting within gene co–expression networks. The Dynamic Tree Cut method is used to divide modules based on clustering results. Merged Dynamic further divides merged modules with similar expression patterns based on module similarity. Subsequent analyses will be conducted on these merged modules. (**B**) Association analysis of gene co–expression network modules with samples.

**Figure 10 ijms-25-13159-f010:**
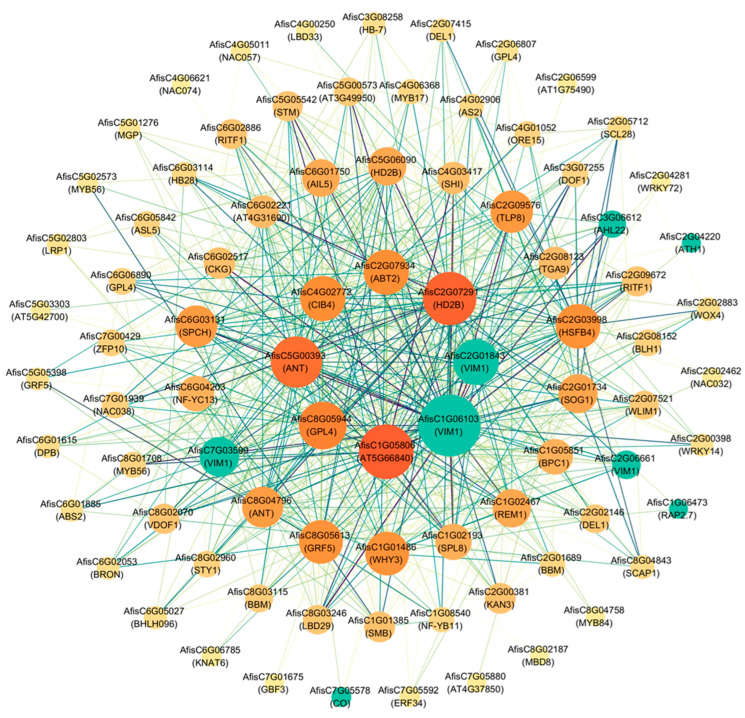
The Welsh onion flowering regulatory candidate genes and key transcription factors co–expression network is visualized within the blue module. Green nodes represent flowering regulatory genes, while all other nodes represent transcription factors. The depth of color of nodes indicates connectivity, and the depth of line color indicates the weight value, with a weight of at least 0.4.

## Data Availability

The original contributions presented in the study are included in the article/Appendix A, further inquiries can be directed to the corresponding authors.

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
