# Peer review of "Transcriptome Profiling Reveals Key Regulatory Networks for Age–Dependent Vernalization in Welsh Onion (Allium fistulosum L.)"

_ijms, 2024, doi:10.3390/ijms252313159_

Round 1

Reviewer 1 Report

Comments and Suggestions for Authors

Dear authors,

The manuscript submitted was broken.

Please fix it and re-submit again.

Author Response

Comments 1: The manuscript submitted was broken. Please fix it and re-submit again.

Responses 1: Thank you for your review. We sincerely apologize for the damage to the Figures in our manuscript. Upon careful inspection, we have resubmitted the manuscript with intact Figures, such as Figure 1 (Now Line 119), which is no longer corrupted. We hope this addresses your concerns and look forward to your further evaluation and comments.

Reviewer 2 Report

Comments and Suggestions for Authors

The paper mainly provide information that a significant correlation between Welsh onion seedling age and vernalization, and transcriptome analysis have potential significance in the age-dependent vernalization with various kinds of seedling ages. The results were very interesting but, some of questions and revision points are exist as follows.

*In order to improve the readability of the paper, it is necessary to simplify or recheck some of sentence structure, please.

-Lines 21 to 23:

-Lines 96 to 99:

-Lines 415 to 418:

-Lines 473 to 478:

*How many plants were used as control? (Line 90, Line 93)

*Cold treatment references (or base) are needed at vernalization (Line 88)

*Humidity condition of indoor (Line 84, Line 94)

*All of DNA’s name are expressed Italic characters.

 *Other minor revision is checked on the manuscript paper.

Comments on the Quality of English Language

Extensive editing of English language required.

Author Response

Comments 1: In order to improve the readability of the paper, it is necessary to simplify or recheck some of sentence structure, please.

-Lines 21 to 23:

-Lines 96 to 99:

-Lines 415 to 418:

-Lines 473 to 478:

Responses 1: Thank you for your review. Regarding the inappropriate sentence structures identified in the article, we have made the necessary corrections to ensure the narrative flow. Naturally, these issues may no longer be in their original locations, as we have adjusted the manuscript's order to align with the journal's specifications. Below, I will outline the current positions of the revised sections:

Original Lines 21 to 23 are now Lines 18 to 20:” Transcriptome analysis led to the identification of genes ~.”

Original Lines 96 to 99 are now Lines 452 to 454:” Three biological replicate samples ~.”;” ~, and by the age of seedlings ~.”

Original Lines 415 to 418 are now Lines 361 to 363:” ~ and the vernalization pathway based on the expression patterns ~.”

Original Lines 473 to 478 are now Lines 429 to 434:” It is critical to point out that genes down-regulated with age, including the VIM1 homologous genes AfisC1G06103, AfisC2G01843, AfisC2G06661, and AfisC7G03599, the RAP2.7 homologous genes AfisC1G06473 and AfisC5G04457, the ATH1 homologous gene AfisC2G04220, and the AHL22 homologous gene AfisC3G06612, are likely in-volved in the regulation of the timing of bolting in Welsh onions.”

Comments 2: How many plants were used as control? (Line 90, Line 93)

Responses 2: More than nine Welsh onions were left untreated as a control (Now Line 445, Lines 448 to 449).

Comments 3: Cold treatment references (or base) are needed at vernalization. (Line 88)

Responses 3: Agree. We have corrected this. “In the Group A trial, when Welsh onions reached 5, 8, and 11 weeks old, they were transferred ~ (Now Lines 442 to 443).”

Comments 4:  Humidity condition of indoor. (Line 84, Line 94)

Responses 4: The humidity in the cultivation room and chamber was maintained at 60±5% (Now Line 440, Line 444).

Comments 5: All DNA names are expressed in italic characters.

Responses 5: Agree. We have corrected this (Now Lines 393 to 394).

Comments 6: Other minor revisions have been checked on the manuscript.

Responses 6: We have thoroughly reviewed the manuscript and corrected all minor issues you pointed out, as well as any other internal issues. Additionally, to accurately present our results, we have added more precise descriptions, such as "the total bolting time for fully vernalized Welsh onions was shortest in the N8V6 treatment" highlighting that the N8V6 treatment had the shortest total bolting time among fully vernalized treatments, not the partially vernalized N5V3 treatment (Now Lines 113 to 114). These are the revisions we have made in response to your comments. We appreciate your efforts and time once again and look forward to your reply.

Reviewer 3 Report

Comments and Suggestions for Authors

Title is correct.

Abstract is written properly. The work is not very revealing, but carried out correctly in terms of method.

This study provides a set of reliable tools for accelerated breeding of Welsh onions, as well as a solid foundation for studying the molecular mechanism of vernalization in Welsh onions.

However, Authors should indicate the aims of the research in the Introduction, without warning what they found in the research.

Line 163-165. That is the conclusion. The problem, however, is that there is nothing particularly revelatory about it, every vegetable gardener knows it. you have to approach the problem differently.

The discussion could have been a little more elaborate. I feel somewhat unsatisfied with the issues discussed.

I would suggest, however, for the sake of clarity, making a Conclusions chapter.

Comments on the Quality of English Language

It's generally ok.

Author Response

Comments 1: However, Authors should indicate the aims of the research in the Introduction, without warning what they found in the research.

Responses 1: Thank you for your review and compliments. We have made appropriate modifications to address the shortcomings you pointed out. We have added a clear statement of our research objectives at the end of the third paragraph in the introduction (Now Lines 67 to 71).

Comments 2: Line 163-165. That is the conclusion. The problem, however, is that there is nothing particularly revelatory about it, every vegetable gardener knows it. you have to approach the problem differently.

Responses 2: We have presented our results with greater precision, avoiding generalizations (Now Lines 104 to 106).

Comments 3: The discussion could have been a little more elaborate. I feel somewhat unsatisfied with the issues discussed.

Responses 3: After thorough discussion, we have changed some of the subheadings in the discussion section to better reflect the content we wish to discuss and made necessary adjustments to the content itself to better showcase our findings (Now Lines 366 to 434).

Comments 4: I would suggest, however, for the sake of clarity, making a Conclusions chapter.

Responses 4: We have added a Conclusions section to the article, which succinctly summarizes the results of this study (Now Lines 496 to 506). Thank you for your review and comments. The above are our responses to your feedback. We appreciate your efforts and time once again and look forward to your reply.

Round 2

Reviewer 1 Report

Comments and Suggestions for Authors

Dear authors

Please find out the attached for the review result.

Author Response

Comments 1: I can’t still understand Fig. 1-A. If authors want to show the difference between bolting and non-bolting, please select CK and N8V6 from G-A and CK and N10V4 from G-B. It provides more clear information to compare.

Responses 1: Thank you for your comments, which are indeed very helpful. Yes, the comparison between N8V6 and N10V4, along with their respective controls, is well-suited for illustrating the state of vernalization in Welsh onions. However, we would like to point out that Welsh onions are plants that exhibit age-dependent vernalization, and the age of the seedlings significantly influences the vernalization process. Therefore, we have included comparisons between N5V3 and N8V3, along with their respective controls, to demonstrate this effect.

Comments 2: However, in order to confirm whether selected genes directly contribute the vernalization, authors should analyze the relative gene expression using qRT-PCR.

Responses 2: Thank you for your suggestion. Regarding the use of qRT-PCR to validate the expression of relevant genes, we offer the following explanations:

  1. All our treatment samples include three biological replicates, with each replicate meeting the sequencing requirements.
  2. We conducted two vernalization experiments at different time points, and the results from these experiments corroborate each other, ensuring the reliability of our research findings.
  3. The current workflows for transcriptome sequencing are highly refined, enabling accurate quantification of the expression levels of each transcript.

Therefore, we believe our data to be highly reliable and accurate, sufficiently supporting our results.

Comments 3: Please confirm the sentence.

Responses 3: Yes, this sentence is an incorrect expression and we have corrected it. “Based on the results of the vernalization test, in Group A, we selected the vernalized non-bolting sample N5V3, the partial-bolting sample N8V3, and the complete-bolting sample N8V6, along with non-vernalized samples before and after each vernalization treatment (N5V0, N8V0, N11V0, and N14V0) for transcriptome construction,” (Now Lines 137 to 141). These are the revisions we have made in response to your comments. We appreciate your efforts and time once again and look forward to your reply.

Reviewer 2 Report

Comments and Suggestions for Authors

Most of revision point was reflected.

Comments on the Quality of English Language

-Minor editing of English language required.

Author Response

Comments 1:

Responses 1: Thank you for your comments, which are indeed very helpful. We have carefully reviewed all the figure captions and have corrected them, “Figure 2. Sample correlation analysis”. Additionally, we have reviewed all capitalization issues within the manuscript and have made the necessary corrections.

Comments 2:

Responses 2: Agree. We have corrected this, “Allium fistulosum L”. Additionally, we have meticulously re-examined the issues within the references section and have corrected them. Thank you for your review and comments. The above are our responses to your comments. We appreciate your efforts and time once again and look forward to your reply.

Round 3

Reviewer 1 Report

Comments and Suggestions for Authors

The current version of a manuscript was improved to be published in IJMS even though it needs some more information to validate the data.